# Artificial Intelligence in Laryngeal Cancer Detection: A Systematic Review and Meta-Analysis

**DOI:** 10.3390/curroncol32060338

**Published:** 2025-06-09

**Authors:** Ali Alabdalhussein, Mohammed Hasan Al-Khafaji, Rusul Al-Busairi, Shahad Al-Dabbagh, Waleed Khan, Fahid Anwar, Taghreed Sami Raheem, Mohammed Elkrim, Raguwinder Bindy Sahota, Manish Mair

**Affiliations:** 1Department of Otolaryngology, University Hospitals of Leicester, Leicester LE1 5WW, UK; mhkhachi@doctors.org.uk (M.H.A.-K.); waleed.a.khan@uhl-tr.nhs.uk (W.K.); mohammed.mohammed21@nhs.net (M.E.); 2Independent Researcher, Leicester LE2 2AD, UK; rusul.albusairi@gmail.com (R.A.-B.); shahad.aldabbagh@doctors.org.uk (S.A.-D.); taghreed.raheem@doctors.org.uk (T.S.R.); 3Department of Maxillofacial Surgery, University Hospitals of Leicester, Leicester LE1 5WW, UK; fahid.anwar@uhl-tr.nhs.uk (F.A.); manish.mair2@nhs.net (M.M.); 4Department of Otolaryngology, University Hospitals of Derby and Burton, Derby DE22 3NE, UK; bindy.sahota@nhs.net

**Keywords:** artificial intelligence (AI), machine learning, laryngeal cancer, laryngoscopy, otolaryngology

## Abstract

(1) Background: The early detection of laryngeal cancer is crucial for achieving superior patient outcomes and preserving laryngeal function. Artificial intelligence (AI) methodologies can expedite the triage of suspicious laryngeal lesions, thereby diminishing the critical timeframe required for clinical intervention. (2) Methods: We included all studies published up to February 2025. We conducted a systematic search across five major databases: MEDLINE, EMCARE, EMBASE, PubMed, and the Cochrane Library. We included 15 studies, with a total of 17,559 patients. A risk of bias assessment was performed using the QUADAS-2 tool. We conducted data synthesis using the Meta Disc 1.4 program. (3) Results: A meta-analysis revealed that AI demonstrated high sensitivity (78%) and specificity (86%), with a Pooled Diagnostic Odds Ratio of 53.77 (95% CI: 27.38 to 105.62) in detecting laryngeal cancer. The subset analysis revealed that CNN-based AI models are superior to non-CNN-based models in image analysis and lesion detection. (4) Conclusions: AI can be used in real-world settings due to its diagnostic accuracy, high sensitivity, and specificity.

## 1. Introduction

The larynx is an important anatomical structure with a role in phonation, respiration, and deglutition. Its physiological function significantly impacts human quality of life by acting as a crucial gateway, directing air into the lungs to breathe and food into the oesophagus on its way to the stomach. As important as its function is, the larynx is highly vulnerable to pathological states; one central malignancy in the upper aerodigestive tract is that of the larynx. It comprises approximately 30–40% of head and neck malignancies and is the most common malignancy in otolaryngology [1]. With the incidence estimated to exceed 24,500 cases per year by 2030 [2], early-stage diagnosis is even more critical to reduce patient mortality and preserve both laryngeal anatomy and vocal fold function [3]. However, laryngeal cancer diagnosis can be challenging. A study published in the Canadian Association of Radiologists Journal identified missed opportunities for an earlier diagnosis of head and neck cancers on prior CT or MRI scans in 4% of cases. Imaging modalities such as ultrasound, computed tomography (CT), and magnetic resonance imaging (MRI) play a crucial role in the staging process and aid in the assessment of tumour size, local invasion, cartilage involvement, and regional lymph node spread [4]. Each tool has its advantages and disadvantages, shaping its uses. The diagnosis of laryngeal lesions begins primarily with indirect laryngoscopy, preferably with endoscopy equipment [5].

Therefore, there is growing interest in utilising artificial intelligence (AI) to enhance clinical outcomes. AI-assisted laryngoscopy offers significant advantages, such as facilitating earlier and more accurate detection of malignant lesions by even non-expert clinicians, which may allow for timely interventions and better prognoses [6].

In this systematic review and meta-analysis, we aim to evaluate artificial intelligence’s diagnostic accuracy and effectiveness in detecting laryngeal cancers by analysing images taken by a laryngeal endoscope during patients’ evaluations due to a laryngeal lesion. Furthermore, we will review limitations such as biases and research gaps.

### The Role of AI in Medical Imaging Analysis

Artificial intelligence (AI) is the simulation of human intelligence in machines that are programmed to think and learn like humans [7]. The role of AI in medicine is constantly expanding. AI promises to revolutionise patient care in the coming years [8].

In his survey, “How Artificial Intelligence Is Shaping Medical Imaging Technology: A Survey of Innovations and Applications” [9], Luis Pinto-Coelho highlighted the current possible applications of AI in medical imaging, such as the interpretation of brain, breast, or other images. He found that the fusion of medical imaging and AI has led to significant advancements in healthcare, from early disease detection to personalised diagnosis and therapy.

Khalifa and Mona, in their review, “AI in Diagnostic Imaging: Revolutionising Accuracy and Efficiency” [10], stated that AI is revolutionising diagnostic imaging, enhancing the accuracy and efficiency of medical image interpretation, and significantly impacting healthcare delivery [10].

AI remains a controversial subject, particularly concerning safety and confidentiality. However, these challenges can be managed by implementing basic safeguards. The above two papers underscore the transformative impact of AI on diagnostic imaging, enhancing accuracy, efficiency, and personalisation in patient care. As AI technologies evolve, their integration into clinical workflows is poised to become a cornerstone of modern medicine, shaping the future of diagnostics and treatment planning.

## 2. Materials and Methods

### 2.1. Preregistration

This review was registered in PROSPERO with ID CRD420250656619 on 21 March 2025.

### 2.2. Search Strategy and Selection Criteria

We followed the PRISMA (Preferred Reporting Items for Systematic Reviews and Meta-Analyses) guidelines in conducting this systematic review and meta-analysis. A careful search was performed across five databases: MEDLINE, EMBASE, EMCARE, PubMed, and the Cochrane Library(Figure 1). The search was conducted to identify studies that used AI with different architectures and tested their sensitivity, specificity, and diagnostic accuracy in detecting laryngeal cancer. The search period was from 1 January 2000 to 1 February 2025. A study was excluded if it was not in English, was a case report, was a systematic review or meta-analysis, had insufficient data, or did not involve cancer detection through imaging. We included original research focused on cancer detection using laryngoscope images. All the included patients were clinically suspicious, and no radiologically suspicious cases were included. The included studies were written in English and had sufficient data for meta-analysis. We formed our search strategy using Boolean operators. The search strategy combined three core concepts: (1) laryngeal cancer, using the terms “laryngeal cancer”, “laryngeal carcinoma”, and “cancer of the larynx”; (2) artificial intelligence, using terms such as “artificial Intelligence,” “AI,” “machine learning,” “deep learning”, “neural network”, “CNN”, and “computer-aided diagnosis”; and (3) diagnostic application, incorporating keywords like “diagnosis”, “detection”, and “classification”. These were further combined with modality-specific terms such as “endoscopy”, “laryngoscopy”, “medical imaging”, “image analysis”, “video endoscopy”, “voice signal”, and “narrowband imaging”. The final Boolean search query used was (“laryngeal cancer” OR “laryngeal carcinoma” OR “cancer of the larynx”) AND (“artificial Intelligence” OR “AI” OR “machine learning” OR “deep learning” OR “neural network” OR “CNN” OR “computer-aided diagnosis”) AND (“diagnosis” OR “detection” OR “classification”) AND (“endoscopy” OR “laryngoscopy” OR “medical imaging” OR “image analysis” OR “video endoscopy” OR “voice signal” OR “narrowband imaging”).

Two independent reviewers (AA and RB) screened all titles and abstracts identified through the database search to determine whether the studies met the inclusion criteria. Full-text articles were retrieved for studies deemed potentially eligible. Both reviewers assessed each full-text report independently against the predefined inclusion and exclusion criteria. Any disagreements were resolved through discussion and, if necessary, consultation with a third reviewer (MM) for an opinion. No automation tools were used in the screening or selection process.

### 2.3. Data Extraction and Analysis

For eligible studies, we extracted data on sensitivity, specificity, sample size, accuracy, and complication rates (Table 1). Studies included in the review compared the diagnostic performance of various machine learning algorithms, including logistic regression, decision trees, and neural networks, in identifying laryngeal cancer. Each study had to report quantitative metrics, such as sensitivity, specificity, accuracy, and the area under the receiver operating characteristic (ROC) curve. We excluded case reports and studies lacking reported sensitivity or specificity values. The risk of bias in the included studies was assessed using the QUADAS-2 (Quality Assessment of Diagnostic Accuracy Studies-2) tool, which evaluates four key domains: patient selection, index test, reference standard, and flow and timing. Two reviewers (AA and RB) assessed each study independently using the QUADAS-2 criteria, and their assessments were compared to ensure consistency. Any discrepancies were resolved through discussion or, if necessary, consultation with a third reviewer (MM). The results of the quality assessment are illustrated in Figure 2. No automated tools were used in the bias assessment process.

### 2.4. Statistical Analysis and Synthesis Methods

To determine eligibility for synthesis, we first tabulated the key characteristics of each included study, including study design, type of AI model used, imaging modality (e.g., laryngeal endoscopy), target condition (laryngeal cancer), and reported diagnostic performance metrics. These characteristics were compared against the predefined inclusion criteria and the planned outcomes for synthesis.

Only studies that evaluated the use of artificial intelligence for detecting laryngeal cancer based on imaging data and reported or allowed for calculation of sensitivity and specificity were included in the quantitative synthesis. Studies that did not provide sufficient diagnostic data or focused on non-cancerous laryngeal conditions were excluded from the meta-analysis but were described narratively when relevant. Two reviewers (AA and RB) carried out this process independently, and any disagreements were resolved by discussion.

We used the Meta-Disc 1.4 software to compute the overall pooled sensitivity, specificity values, and diagnostic accuracy and to generate forest plots for these values. Two reviewers (AA and RB) independently extracted data using a standardised data collection form. For each included study, relevant information such as study design, population characteristics, AI model details, and diagnostic performance metrics was recorded. In cases where true positive, true negative, false positive, and false negative values were not explicitly reported, these values were calculated based on the available data, including sample size, number of cancer cases, and reported sensitivity and specificity. No automation tools were used in the data extraction process. When necessary, efforts were made to contact study authors to clarify or obtain missing data. Discrepancies between reviewers were resolved through discussion and consensus. Forest plots were generated to illustrate the variability in sensitivity and specificity estimates, accompanied by 95% confidence intervals (CIS) (Figure 3).

## 3. Results

We carried out a thorough search across five databases, including MEDLINE [25 records], EMBASE (28 records), EMCARE (11 records), PubMed (26 records), and Cochrane Library (1 record), yielding a total of 91 records. After removing 36 duplicates and 32 studies from the title, 23 studies remained for eligibility screening. Of these, eight were excluded from the title, leaving 15 studies on the final screen and included in the final analysis. The complete search process is outlined in the PRISMA flow diagram (Figure 1).

We excluded eight studies due to insufficient data [24,26,27,28,29,30,31,32].

All included studies [11,12,13,14,15,16,17,18,19,20,21,22,23,24,25] were retrospective experimental studies, were published in English, and provided sufficient data to calculate true positive, true negative, false positive, and false negative rates. These studies focused on detecting laryngeal cancer through laryngoscope videos or photos and comparing the effectiveness of AI in distinguishing cancerous lesions from benign ones. The majority of studies utilised flexible nasoendoscopy, with only one study employing rigid endoscopy. Regarding the imaging modalities used in the included studies, twelve employed white light imaging (WLI), two utilised a combination of WLI and narrow band imaging (NBI), and one study used NBI alone.

The data collected are presented in Table 1.

### 3.1. Overall Pooled Analysis

This review was conducted on 15 studies, including 22,842 images of the larynx taken from 13,570 patients. The analysis yielded an overall pooled sensitivity of 78% (95% CI: 77–78%), showing a strong ability to correctly identify negative cases. The pooled specificity was found to be 86% (95% CI: 86–87%), indicating strong performance in correctly identifying negative cases. The diagnostic odds ratio (DOR), a global measure of test effectiveness, resulted in a pooled value of 53.77 (95% CI: 27.38–105.62), suggesting consistent overall discriminatory capacity. The SROC curve, which produced an AUC of 0.9380 and a Q index of 0.8750, emphasising excellent overall diagnostic accuracy of the AI-based models under evaluation. The results are illustrated in Figure 3.

### 3.2. Subset Analysis: Diagnostic Accuracy of CNN-Based MODELS vs. Non-CNN Models

We conducted a subset analysis to compare the diagnostic accuracy of Convolutional Neural Network (CNN)-based models with that of non-CNN-based models used in our included studies. The presented meta-analysis compares the diagnostic performance of non-CNN-based and CNN-based models using several forest plots and SROC curves. The non-CNN group yielded a pooled sensitivity of 0.65 (95% CI: 0.64–0.66) and specificity of 0.80 (95% CI: 0.79–0.81), with high heterogeneity (I^2^ = 96.7% and 97.9%, respectively). The corresponding symmetric SROC curve showed an AUC of 0.9044, indicating good overall accuracy (Figure 4). In contrast, CNN-based models demonstrated superior diagnostic accuracy, with a pooled sensitivity of 0.85 (95% CI: 0.84–0.86) and specificity of 0.90 (95% CI: 0.89–0.90), both with high but slightly lower heterogeneity compared to non-CNN models. The AUC of the SROC for CNN-based models was 0.9307, suggesting enhanced discriminative ability. Furthermore, the diagnostic odds ratio (DOR) was markedly higher in the CNN group (pooled DOR = 51.04, 95% CI: 37.53–69.41) than in the non-CNN group (pooled DOR = 15.13, 95% CI: 10.09–22.67), emphasising the significant improvement in diagnostic performance with CNN-based approaches (Figure 5). Overall, the results support the superior accuracy and diagnostic strength of CNN-based models in medical image classification tasks.

## 4. Discussion

The rapid development of machine learning and artificial intelligence with potent analytical abilities has raised several concerns and questions: will AI imminently replace medical expertise? [33]. The answer to this question can be unpredictable; however, it remains a valuable tool for improving our practice. In their article, Davenport T. and Kalakota R. explain the implications of AI for the healthcare workforce and how it can improve, but not replace, medical practice. Healthcare jobs most likely to be affected by AI will be those that involve dealing with digital information, such as radiology and pathology, for example [34]. Among AI’s most prominent and promising uses is in radiology, where deep learning excels in lesion detection and medical image analysis [35]. Aside from radiology, comparable developments are being seen with the application of AI in histopathology, where AI has the potential to alleviate the workload of human experts, make reporting more consistent and unbiased, and promote better clinical outcomes [36]. Our findings further support these developments by demonstrating that AI can accurately assist in diagnosing laryngeal cancer and reduce the proportion of missed cases due to a lack of experience. Aside from greater diagnostic accuracy, AI reduces delays in analysing images and videos by automating image interpretation, triaging suspicious lesions faster. For laryngeal cancer, where early-stage conditions are often more suitable for organ-preserving therapies like transoral laser microsurgery or radiotherapy, early diagnosis significantly influences treatment choices and impacts long-term voice and airway health. AI also helps in standardising terminology used during image analysis. By eliminating inter-observer variability, AI systems can provide more uniform care among institutions and geographic centres.

### 4.1. Limitations

This meta-analysis has several limitations that warrant consideration. The studies in the analysis were very heterogeneous regarding the sources, image quality, and deep learning algorithms, thus introducing selection bias into the study. Improved study designs utilising standardised modality images and specific deep-learning models must be employed in future work to improve reliability. Also, most included studies lacked external validation datasets, which hindered generalisability in real-world practice settings. There is a clear need for high-quality prospective studies with comprehensive external validation cohorts to address this gap and enhance the current evidence base. Finally, while many papers demonstrated the superior diagnostic performance of deep learning models compared to clinicians, particularly those with less experience, insufficient data were provided on performance by clinicians with and without AI support, making an aggregated objective appraisal difficult. Furthermore, the majority of these studies focused on Chinese populations. Therefore, there is an increased need to include data on ethnically and geographically distinct populations to maximise the applicability and generalisability of the findings.

### 4.2. Future Application

The evidence presented in this review elucidates the exceptional diagnostic performance of artificial intelligence (AI) models in the identification of laryngeal lesions, with individual studies demonstrating performance levels that are equal to or exceed those of experienced clinicians. Consequently, this warrants the integration of AI software to support routine otolaryngology clinics, particularly in settings with limited resources or areas lacking specialised expertise. AI-assisted laryngoscopy has the potential to facilitate earlier diagnoses of malignancies, standardise assessments, and minimise diagnostic delays. Nonetheless, it is imperative for clinicians to acknowledge the limitations of these models and to provide appropriate clinical oversight to mitigate the risk of over-reliance on such technologies.

## 5. Conclusions

We have found that AI has excellent diagnostic accuracy, with high sensitivity and specificity, and can be used in real-world settings and otolaryngology clinics. Further improvement is still a great opportunity. According to our review, research with high-image-number data has better outcomes. This leads us to the suggestion of having one centralised worldwide database centre for images of laryngeal lesions where AI can be trained for better outcomes.

## Figures and Tables

**Figure 1 curroncol-32-00338-f001:**
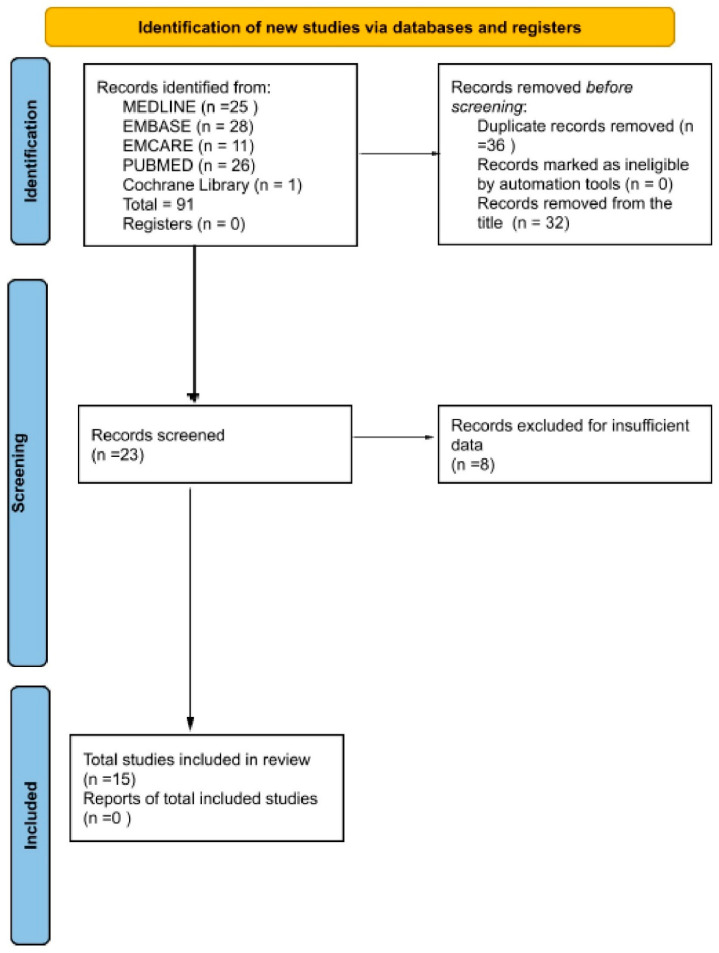
PRISMA flow diagram.

**Figure 2 curroncol-32-00338-f002:**
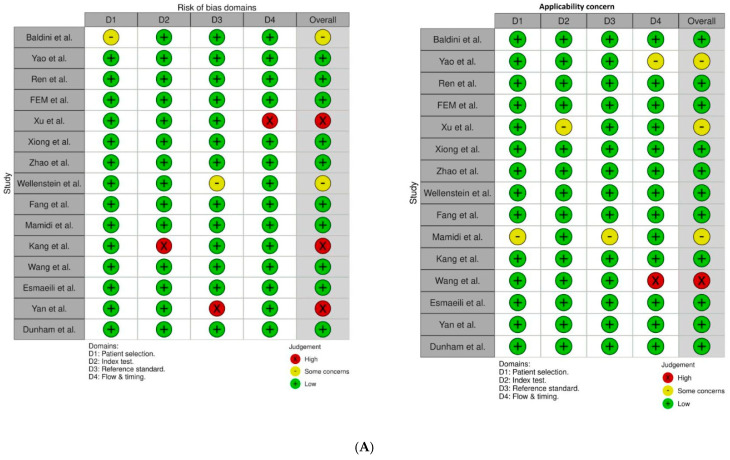
(**A**). Risk of bias and applicability concern summary [11,12,13,14,15,16,17,18,19,20,21,22,23,24,25] and (**B**). risk of bias and applicability concerns graphs.

**Figure 3 curroncol-32-00338-f003:**
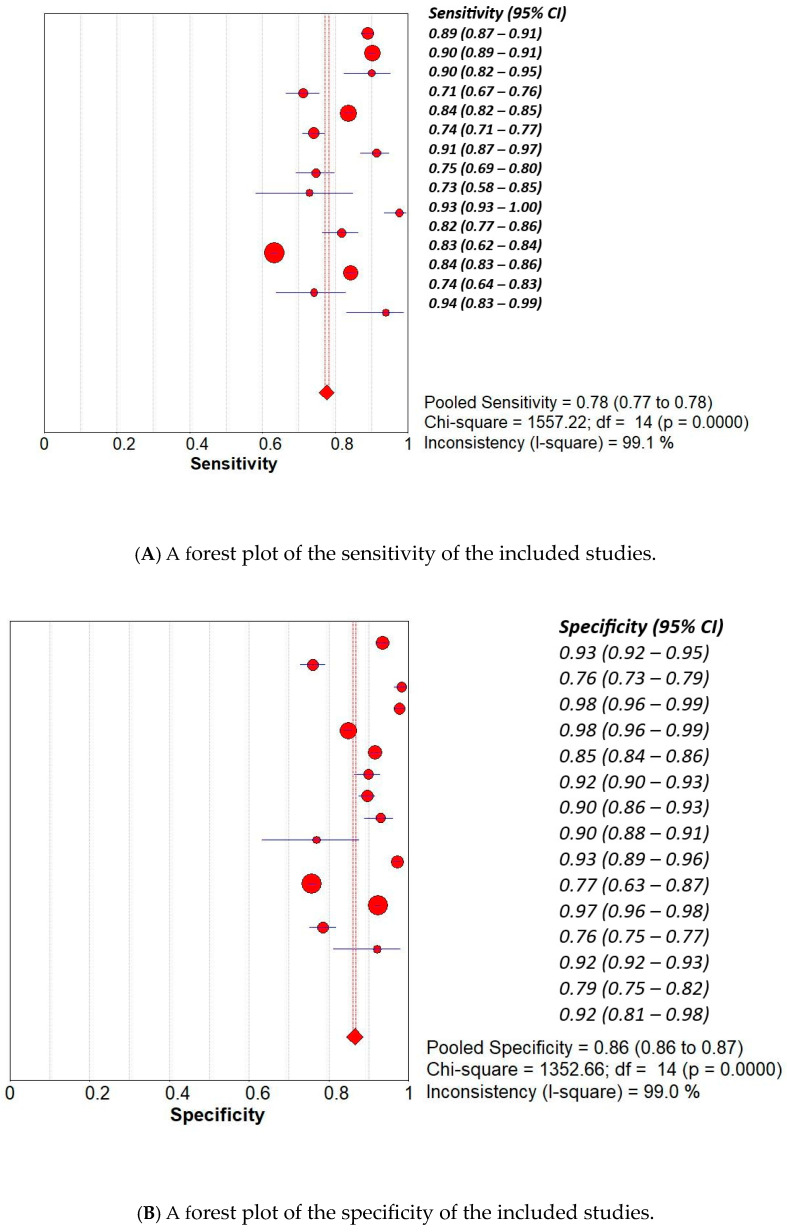
The (**A**)—sensitivity, (**B**)—specificity, (**C**)—SROC, and (**D**)—diagnostic accuracy of included studies.

**Figure 4 curroncol-32-00338-f004:**
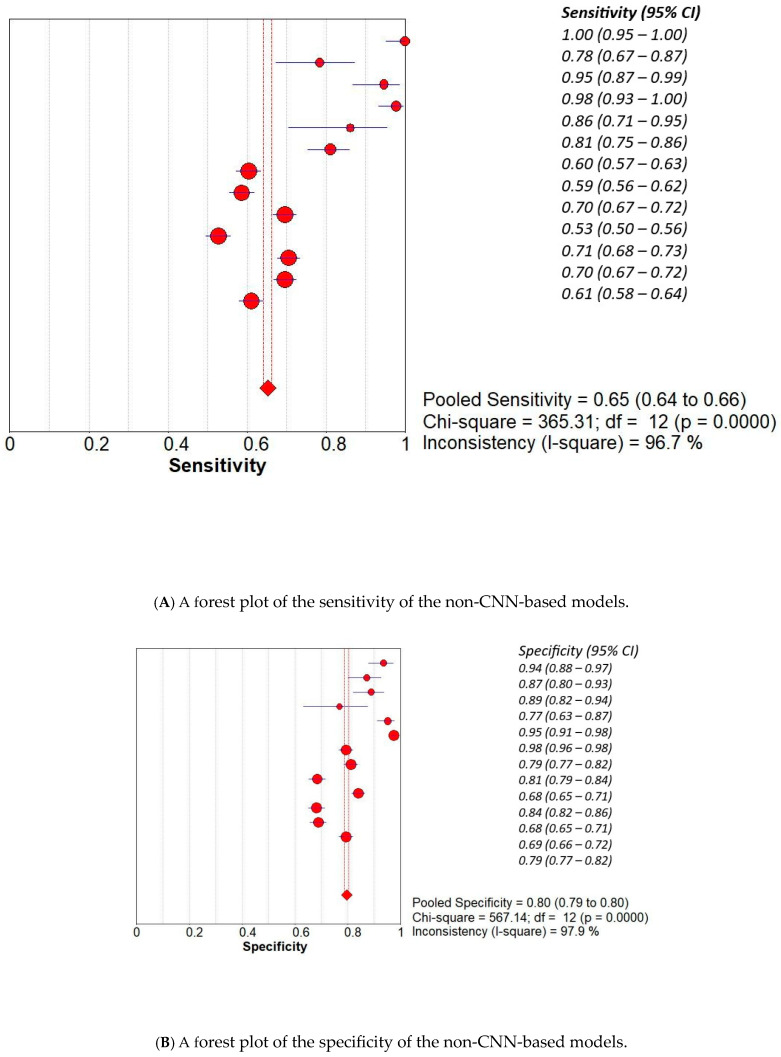
The (**A**)—sensitivity, (**B**)—specificity, (**C**)—SROC, and (**D**)—diagnostic accuracy of non-CNN-based models.

**Figure 5 curroncol-32-00338-f005:**
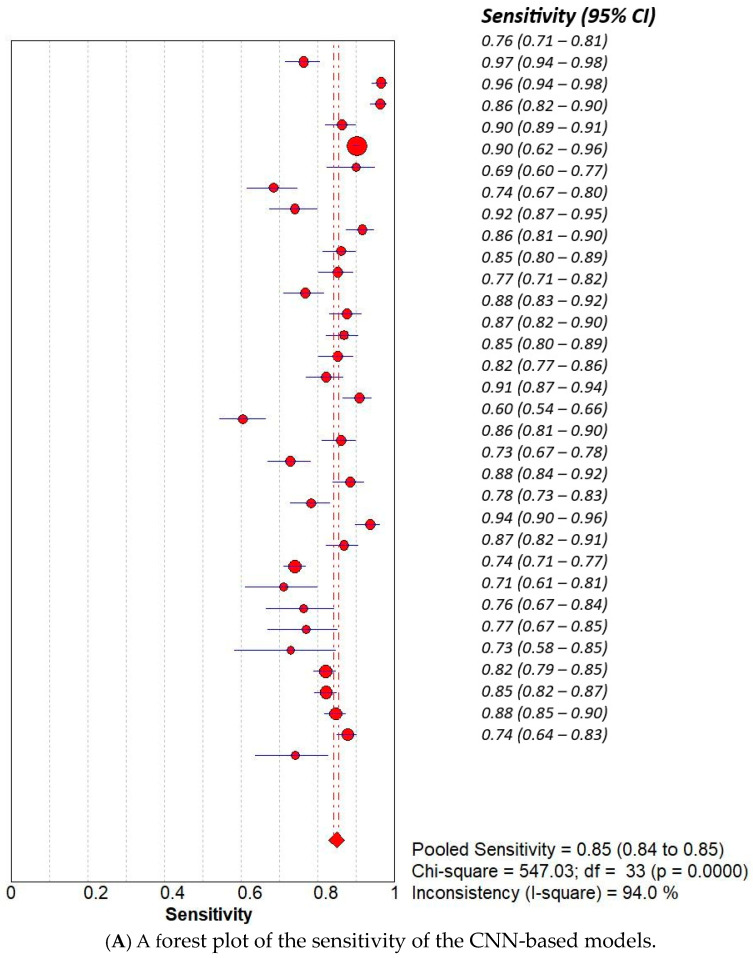
The (**A**)—sensitivity, (**B**)—specificity, (**C**)—SROC, and (**D**)—diagnostic accuracy of CNN-based models.

**Table 1 curroncol-32-00338-t001:** Summary of data collected from involved studies.

Study	AI Model	TP	TN	FP	FN
Baldini et al. [11]	Int shallow CNN	264	313	45	82
	INT ResNet-50	334	339	19	12
	INT MobileNetv2	333	339	19	13
	EXT ResNet-50	272	331	9	43
Yao et al. [12]	CNN	3277	612	193	356
Ren et al. [13]	CNN	90	393	7	10
Lee et al. [14]	YOLOV5	137	392	8	63
	YOLOV6	148	390	10	52
Xu et al. [15]	Densenet201 INTERNAL	230	220	18	21
	Densenet201 EXTERNAL	222	230	36	36
	Alexnet INTERNAL	214	194	43	37
	Alexnet EXTERNAL	198	201	64	60
	Inception v3 INTERNAL	220	213	24	31
	Inception v3 EXTERNAL	224	189	76	34
	Mnasnet INTERNAL	214	231	7	37
	Mnasnet EXTERNAL	212	263	2	46
	Mobilenet v3 INTERNAL	228	132	105	23
	Mobilenet v3 EXTERNAL	156	212	53	102
	Resnet152 INTERNAL	216	217	20	35
	Resnet152 EXTERNAL	188	248	18	70
	Squeezenet1 INTERNAL	222	207	30	29
	Squeezenet1 EXTERNAL	202	212	53	56
	Vgg19 INTERNAL	235	207	30	16
	Vgg19 EXTERNAL	224	243	22	34
Xiong et al. [16]	DCNN	628	1815	166	220
Zhao et al. [17]	RF	74	118	8	0
	DV	58	110	16	16
	SVM	70	112	14	4
Wellenstein et al. [18]	YOLOv5s	69	303	23	28
	YOLOv5m	74	284	42	23
	YOLOv5sl	70	295	37	21
Fang et al. [19]	Faster R-CNN	35	213	16	13
Mamidi et al. [20]	[Vit]	127	40	12	3
Kang et al. [21]	ILCDS ex	31	187	9.47	5
	ILCDS in	184	979	23.97	43
Wang et al. [22]	LR	627	723	187.46	411
	SVM	609	740	170.17	429
	RandomForest	722	623	286.65	316
	ExtraTrees	548	765	144.69	490
	XGBoost	733	621	289.38	305
	LightGBM	723	627	283.01	315
	MLP	634	723	187.46	404
Esmaeili et al. [23]	DenseNet121	563	1383	152	123
	EfcientNetB0V2	564	1386	149	122
	ResNet50V2	581	1434	101	105
	Ensemble model.	602	1461	74	84
Yan et al. [24]	R-CNNs	66	503	137	23
Dunham et al. [25].	CNN	46	47	4	3

TP: True positive, TN: true negative, FP: false positive, FN: false negative.

## Data Availability

All available data included in the main manuscript and no more raw data.

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
