# Peer review of "Artificial Intelligence in Laryngeal Cancer Detection: A Systematic Review and Meta-Analysis"

_curroncol, 2025, doi:10.3390/curroncol32060338_

Round 1
Reviewer 1 Report
Comments and Suggestions for Authors
For the introduction:
What would you desire from AI-assisted laryngoscopy? Pure diagnostic or influencing therapeutic strategies? Higher accuracy in detecting (pre-)malignant lesions, compared to (less-)experienced clinicians? Earlier diagnosis? Reduction of biopsies? Reduction of diagnostic (laser-)excisions. More accurate determination of extension? more accurate staging? Earlier detection of residual or recurrences?
Discussion:
Something I don't understand: line 238-241 "AI can substantially reduce delays in analysing images and videos, enabling quicker clinical decisions and potentially earlier intervention. This is especially beneficial in conditions where timely diagnosis directly influences patient outcomes."
Author Response
Comment 1: What would you desire from AI-assisted laryngoscopy? Pure diagnostic or influencing therapeutic strategies? Higher accuracy in detecting (pre-)malignant lesions, compared to (less-)experienced clinicians? Earlier diagnosis? Reduction of biopsies? Reduction of diagnostic (laser-)excisions. More accurate determination of extension? More accurate staging? Earlier detection of residual or recurrences?
Response 1: Thank you for this valuable comment; we have addressed this point as follows
The diagnosis of laryngeal cancer can be challenging, particularly in its early or precancerous stages; there is growing interest in utilising artificial Intelligence (AI) to enhance clinical outcomes. AI-assisted laryngoscopy offers significant advantages, such as facilitating earlier and more accurate detection of malignant lesions by even non-expert clinicians, which may allow for timely interventions and better prognoses. You can find this in the PDF version, page 2, under the introduction section, line 51-54
Comment 2:
Discussion:
Something I don't understand: line 238-241 "AI can substantially reduce delays in analysing images and videos, enabling quicker clinical decisions and potentially earlier intervention. This is especially beneficial in conditions where timely diagnosis directly influences patient outcomes."
Response 2: Thank you for this valuable comment; we have addressed this point as follows
AI reduces delays in analysing images and videos by automating image interpretation and triaging suspicious lesions faster. For laryngeal cancer, where early-stage conditions are often more suitable for organ-preserving therapies like transoral laser microsurgery or radiotherapy, early diagnosis significantly influences treatment choices and impacts long-term voice and airway health. You can find this in the PDF version, page 15-16, under the Discussion section, line 260-265
Reviewer 2 Report
Comments and Suggestions for Authors
I have reviewed this interesting systematic review to assess the artificial intelligence in detecting laryngeal cancer by laryngeal endoscopy.
Abstract: Instead of background, the authors only presented the aims of this study. The Methods are too detailed, including citations of Table 1 and Figure 1.
Introduction: Methods like ultrasound, CT and MRI scans are mainly employed for staging the disease instead of for its diagnosis. Please improve the elaboration of these concepts.
Methods: Table 1 could be replaced by a paragraph with the Boolean search description.
I appreciated that the authors mentioned the limitations of thsi review. In fact, heterogeneity and different image quality is an important aspect.
Author Response
Comment 1: Abstract: Instead of background, the authors only presented the aims of this study. The Methods are too detailed, including citations of Table 1 and Figure 1
Response 1: Thank you for this valuable comment; we have addressed this point as follows
Abstract(1) Background: Early detection of laryngeal cancer is crucial for achieving superior patient outcomes and preserving laryngeal function. Artificial Intelligence (AI) methodologies can expedite the triage of suspicious laryngeal lesions, thereby diminishing the critical timeframe required for clinical intervention. (2) Methods: We included all studies published up to February 2025. We conducted a systematic search across five major databases: MEDLINE, EMCARE, EMBASE, PubMed, and the Cochrane Library. We included 15 studies, with a total of 17,559 patients. A risk of bias assessment was performed using the QUADAS-2 tool. We conducted data synthesis using the Meta Disc 1.4 program. You can find this in the PDF version, page 1 lines 17-24
Comment 2: Introduction: Methods like ultrasound, CT and MRI scans are mainly employed for staging the disease instead of for its diagnosis. Please improve the elaboration of these concepts.
Response 2: Thank you for this valuable comment; we have addressed this point as follows
A study published in the Canadian Association of Radiologists Journal identified missed opportunities for earlier diagnosis of head and neck cancers on prior CT or MRI scans in 4% of cases. Imaging modalities such as ultrasound, computed tomography (CT), and magnetic resonance imaging (MRI) play a crucial role in the staging process and aid in the assessment of tumour size, local invasion, cartilage involvement, and regional lymph node spread[4].
You can find this in the PDF version, page 1, lines 42-47
Comment 3: Methods: Table 1 could be replaced by a paragraph with the Boolean search description.
Response 3: Thank you for this valuable comment; we have addressed this point as follows
The search strategy combined three core concepts: (1) laryngeal cancer, using the terms “laryngeal cancer,” “laryngeal carcinoma,” and “cancer of the larynx”; (2) artificial Intelligence, using terms such as “artificial Intelligence,” “AI,” “machine learning,” “deep learning,” “neural network,” “CNN,” and “computer-aided diagnosis”; and (3) diagnostic application, incorporating keywords like “diagnosis,” “detection,” and “classification.” These were further combined with modality-specific terms such as “endoscopy,” “laryngoscopy,” “medical imaging,” “image analysis,” “video endoscopy,” “voice signal,” and “narrowband imaging.” The final Boolean search query used was: (“laryngeal cancer” OR “laryngeal carcinoma” OR “cancer of the larynx”) AND (“artificial Intelligence” OR “AI” OR “machine learning” OR “deep learning” OR “neural network” OR “CNN” OR “computer-aided diagnosis”) AND (“diagnosis” OR “detection” OR “classification”) AND (“endoscopy” OR “laryngoscopy” OR “medical imaging” OR “image analysis” OR “video endoscopy” OR “voice signal” OR “narrowband imaging”). You can find this in the PDF version, page 3, lines 96-109
Comment 4: I appreciated that the authors mentioned the limitations of this review. In fact, heterogeneity and different image quality are important aspects.
We appreciate your positive feedback and for acknowledging our discussion of the review's limitations. We agree that aspects such as heterogeneity and varying image quality are essential considerations.
Reviewer 3 Report
Comments and Suggestions for Authors
Thank you for the opportunity to review this timely and relevant systematic review exploring the diagnostic performance of AI in detecting laryngeal cancer through endoscopic images. The field of AI-driven diagnostics is rapidly expanding, and this study makes a valuable contribution by synthesizing available evidence. However, I would like to raise several concerns and suggestions that should be addressed prior to publication.
Major Comments:
1.Lack of Clarity Regarding Study Populations
The included studies primarily focus on images acquired from patients with suspected laryngeal pathology. However, the review lacks clarity on whether these populations consist solely of patients with clinically or radiologically suspicious lesions or whether asymptomatic controls were also included. This is a critical issue, as diagnostic accuracy can vary significantly depending on disease prevalence and clinical setting. Please specify:
Were the non-cancer cases in each study asymptomatic individuals, patients with benign lesions, or both?
2.Insufficient Information on Imaging Modalities Used
The manuscript refers broadly to “laryngoscopic images” without distinguishing between key modalities such as: Rigid endoscopy vs. flexible laryngoscopy, White light imaging (WLI) vs. narrow band imaging (NBI), Standard endoscopy vs. stroboscopy or contact endoscopy
Since image quality and content vary considerably between these modalities, their diagnostic value for AI can also differ. It would strengthen the review to clearly:
Provide a table or supplemental summary describing the imaging modality used in each included study.
Stratify diagnostic accuracy by imaging modality if possible (e.g., subgroup analysis).
3.Clinical Heterogeneity and External Validity
While the authors mention the lack of external validation as a limitation, the degree of clinical heterogeneity among the included studies is not fully explored. This includes:
Patient demographics (most studies conducted in Asian populations, especially China)
Variability in lesion stages and types (e.g., glottic vs. supraglottic cancer)
A more detailed breakdown of these factors would help readers interpret the generalizability of the findings. Please consider including a table summarizing the key clinical characteristics of each included dataset if available.
The manuscript presents promising evidence for AI use in laryngeal cancer diagnosis; however, to ensure scientific rigor and clinical relevance, the authors should better clarify the patient selection criteria, imaging modalities used, and population characteristics in the included studies. Additional stratified analyses by modality or diagnostic setting would add value.
Author Response
Comment 1: 1. Lack of Clarity Regarding Study Populations
The included studies primarily focus on images acquired from patients with suspected laryngeal pathology. However, the review lacks clarity on whether these populations consist solely of patients with clinically or radiologically suspicious lesions or whether asymptomatic controls were also included. This is a critical issue, as diagnostic accuracy can vary significantly depending on disease prevalence and clinical setting. Please specify:
Were the non-cancer cases in each study asymptomatic individuals, patients with benign lesions, or both?
Response 1:Thank you for this valuable comment.
All patients included in the studies were clinically suspected of having laryngeal pathology; asymptomatic individuals were excluded from the study populations. The non-cancer cases comprised patients with benign laryngeal lesions rather than asymptomatic controls. We have clarified this point in the Methods section under 'Inclusion Criteria'. Additionally, in the demographic table provided in the supplementary files, we have detailed the population type included in each study. However, it is essential to note that our review focused exclusively on cancer and precancerous lesions. You can find this in the PDF version, page 2, Lines 93-95 and in the supplementary files.
Comment 2: 2. Insufficient Information on Imaging Modalities Used
The manuscript refers broadly to “laryngoscopic images” without distinguishing between key modalities such as: Rigid endoscopy vs. flexible laryngoscopy, White light imaging (WLI) vs. narrow band imaging (NBI), Standard endoscopy vs. stroboscopy or contact endoscopy.
Since image quality and content vary considerably between these modalities, their diagnostic value for AI can also differ. It would strengthen the review to clearly:
Provide a table or supplemental summary describing the imaging modality used in each included study.
Stratify diagnostic accuracy by imaging modality if possible (e.g., subgroup analysis).
Response 2: Thank you for this important observation.
We have now included details of the imaging modalities used in each study within the demographic table provided in the supplementary materials. The majority of studies utilised flexible nasoendoscopy, with only one study employing rigid endoscopy. Due to this limited variability, a meaningful subgroup analysis based on endoscopic modality is not feasible. Similarly, while most studies used white light imaging (WLI), the use of advanced modalities such as narrow band imaging (NBI), stroboscopy, or contact endoscopy was rare, further limiting the potential for stratified diagnostic accuracy analyses. You can find this in the PDF version, page 4, lines 173-177 and in the supplementary files.
Comment 3: 3. Clinical Heterogeneity and External Validity
While the authors mention the lack of external validation as a limitation, the degree of clinical heterogeneity among the included studies is not fully explored. This includes:
Patient demographics (most studies conducted in Asian populations, especially China)
Variability in lesion stages and types (e.g., glottic vs. supraglottic cancer)
A more detailed breakdown of these factors would help readers interpret the generalizability of the findings. Please consider including a table that summarises the key clinical characteristics of each included dataset, if available.
The manuscript presents promising evidence for AI use in laryngeal cancer diagnosis; however, to ensure scientific rigor and clinical relevance, the authors should better clarify the patient selection criteria, imaging modalities used, and population characteristics in the included studies. Additional stratified analyses by modality or diagnostic setting would add value.
Response 3: Thank you for this insightful comment.
We agree that clinical heterogeneity is an essential factor influencing the generalizability of findings in AI-based diagnostic research. To address this, we have now included a detailed demographic table in the supplementary materials that summarises the key clinical characteristics of each included study. This table outlines the country of the study population, site of the lesion (noting that all included studies focused on glottic lesions), and the imaging modalities used.
While most studies were conducted in Asian populations, particularly China, we have also included studies from Europe and the United States to enhance the diversity of the dataset. Regarding lesion variability, only studies that focused on glottic lesions were included.
Due to limited heterogeneity in imaging modalities and lesion sites, stratified analyses were not feasible.
For more details, please refer to the supplementary files.
Round 2
Reviewer 2 Report
Comments and Suggestions for Authors
The paper improved after the changes.
Reviewer 3 Report
Comments and Suggestions for Authors
The authors have adequately addressed all reviewer comments and revised the manuscript accordingly. The updated version demonstrates improved clarity, organization, and scientific rigor. I believe the manuscript is now suitable for publication and recommend acceptance in its current form.